# Emissions and Exposures Associated with the Use of an Inconel Powder during Directed Energy Deposition Additive Manufacturing

**DOI:** 10.3390/ijerph20136206

**Published:** 2023-06-22

**Authors:** Marelizé van Ree, Sonette du Preez, Johan L. du Plessis

**Affiliations:** Occupational Hygiene and Health Research Initiative (OHHRI), North-West University, Potchefstroom 2531, South Africa

**Keywords:** 3D printing, metal AM, particle emission rate, occupational exposure, respiratory exposure, health effects

## Abstract

Additive manufacturing (AM) has been linked to potential exposure-related health risks, however, there is a paucity of sufficient research. This study aimed to supply information regarding emissions and exposure during directed energy deposition (DED) AM using inconel 718, with the main constituents being nickel, chromium, and cobalt. By using standardized occupational hygiene methods, the measurement strategy consisted of a combined approach, including powder characterization, particle emission monitoring, and personal exposure monitoring of AM operators. Powder characterization of virgin and used powder indicated no significant difference in particle size, shape, or elemental composition. Particle number emissions ranged between 10^2^ and 10^5^ p/cm^3^ for submicron particles (<1 µm in size). There was no significant difference in the particle emission rate between the three phases of AM or the two types of DED machines (*p* > 0.05). The particle emission rate for submicron particles peaked at 2.8 × 10^9^ p/min. Metals of concern to human health were detected during the AM process but were considerably lower than the relevant exposure limits. This study confirms particle emissions, predominantly in the submicron range, above the background concentration during DED AM and, although insignificant in terms of potential health effects, AM operators are exposed to detectable concentrations of the metal constituents of inconel.

## 1. Introduction

Additive manufacturing (AM) is an innovative technology with the potential of revolutionizing the manufacturing industry [1,2,3]. AM can be described as the creation of any object, no matter how complex, through the layering of materials, having various advantages contributing to the increased interest in this field [1,4,5]. Some of these advantages include producing complex shapes, improving material usage while decreasing waste, and reduction in production costs as well as time [1,6,7,8,9]. Notwithstanding the last-mentioned advantages, there are some uncertainties regarding the emissions and health hazards related to AM as is often the case with new technologies [6]. Despite AM having potential occupational hazards related to the process, there is still insufficient published research on metal AM, especially directed energy deposition (DED), regarding AM operator health risks [9,10].

AM has seven standard process categories which include DED [3]. The DED process makes use of either powder or wire, fed to the substrate by a deposition head while simultaneously being melted by an energy source such as a laser, electron beam, electric, or plasma arc. The feedstock material is selectively deposited by the deposition head to build a 3D object [9,11,12]. In this study, two laser-based DED machines were investigated, namely laser engineered net shaping (LENS) and laser metal deposition (LMD), with the main difference being the LENS machine is enclosed while the LMD is an open-table machine. Furthermore, a nickel-based metal powder, specifically inconel (IN718), was investigated as feedstock material. Inconel is a metal powder with its main constituents being nickel, chromium, and cobalt [13,14]. Nickel-based inconel is classified as a super alloy with high-temperature strength along with corrosion and oxidation resistance [15,16].

The AM process can be divided into three phases with different tasks, emissions, and exposure of AM operators. The phases of AM are pre-processing (designing of object and preparation of machine), processing (printing of object), and post-processing (removal, cleaning, and further processing of object) [8,17,18]. When workers are exposed to chemicals of concern to their health, there are various routes of exposure indicating the way the chemical is absorbed into the body. The size, solubility, and shape of particles determine the route of exposure. Inhalation is the most significant route of exposure for metal-based powders [8]. A particle has to be airborne and emitted or displaced within the breathing zone to be inhaled. Particles are categorized as inhalable or respirable, referring to the area in the lungs where the particles may be deposited [19]. Inhalable particles may be deposited anywhere along the respiratory tract while respirable particles refer to particles penetrating much deeper into the lungs, reaching the alveoli or gas-exchange region [19,20,21].

Few studies on emissions and exposure during the process of AM, along with the potential health implications of AM operators, have been published, with the calculation of the emission rate of particles in metal AM lacking [1,6,8,18,22,23,24,25,26,27]. The main findings of these studies were notable differences between virgin and used metal powders while used powder contained smaller particles in comparison to virgin powder. There is a high particle number concentration inside the chamber during processing (10^6^ p/cm^3^ for particles ranging between 0.04 and 0.11 µm in size), however, with the enclosed machine, the particle number concentration in the area was relatively low (10^4^ p/cm^3^) [1,24]. On the other hand, during open-table AM, the particle number concentration in the area was notably higher (9.3 × 10^5^ p/cm^3^ for particles ranging between 0.01 and 1.09 µm in size). There are clear peaks in particle number concentration during pre- and post-processing tasks associated with manual handling of the powder (10^4^–10^6^ p/cm^3^). Emitted particles (<300 nm in size) have been considerably smaller than that of feedstock powder whilst different phases of AM displayed different peak size ranges [1,6,8,18,24,27].

Although an advanced technology, AM still requires manual input and handling of feedstock material. The process generates varying levels of airborne particles during the three phases, which causes a risk of respiratory exposure. In the case of a nickel-based inconel powder, various respiratory diseases may develop, however, the most imperative hazard that may be associated with this powder is cancer with long-term exposure. According to the International Agency for Research on Cancer (IARC), the main constituents of this powder have classifications ranging from carcinogenic to humans (Group 1—nickel and hexavalent chromium) to possibly carcinogenic to humans (Group 2B—cobalt) which increases the need to control exposure [28,29]. South African legislation requires employers to assess and control exposure to hazardous chemical agents (HCAs) with appropriate measures while providing time-weighted average (TWA) occupational exposure limits (OELs) which may not be exceeded [30]. However, the provided OELs, either for the inhalable or respirable fraction, do not take account of particles in the submicron size range.

Considering that there are known risks to the health of AM operators, but limited research regarding the quantification and control thereof, this research study aimed to provide vital information. The specific objectives of the study are (i) physico-chemical characterizing of inconel powder particles, (ii) assessing the particle number concentration and rate of particle emissions in the work area, as well as (iii) establishing personal respiratory exposure of AM operators to metals of concern to human health at a South African AM facility.

## 2. Materials and Methods

This study was conducted at a South African research facility where the measurement strategy consisted of a combined approach to get a complete overview of emissions and exposure during the AM process. This included powder characterization, particle emission monitoring, and personal exposure monitoring of AM operators. The entire process of DED AM was observed (pre-processing, processing, and post-processing) for three cycles, manufacturing three identical objects using both the enclosed LENS and open-table LMD machines. Through observation and note-taking, tasks performed during each phase as well as the duration of tasks and phases, and any other relevant factors that may have influenced measurements, were identified.

### 2.1. Facility and Process Description

This AM research facility consists of numerous rooms used during the process including a general workshop, an AM laboratory, and lastly, a dedicated area where post-processing tasks take place (see Table 1 for complete facility summary). This study involved two DED machines making use of the LENS and LMD technologies. The LENS (850-R, Optomec, Albuquerque, NM, USA) is an enclosed system with 5-axis motion control and an IRE-Polus Group (IPG) Fiber Laser (1 or 2 kW), used in the general workshop where various other tasks are carried out. The open-table LMD system consists of a Kuka robot (Kuka Robotics, Augsburg, Germany) with a laser deposition head that can be used for laser cladding, laser welding, AM, and refurbishment applications on various surfaces. This system also uses an IPG fiber laser (2 kW) and is situated in an AM laboratory. The post-processing tasks take place in the dedicated area, a room with various types of machinery including a sandblasting station which was the main finishing task. A glass bead abrasive was used with particle size ranging from 45 to 1000 µm. This facility uses AM machines mainly for research purposes. Various metal powders are used with the DED machines, and for the purpose of this study, an inconel powder, IN718 (NI-202-1, Praxair Surface Technologies, Indianapolis, IN, USA) was used. To achieve the objectives of this study and the research conducted by the facility, identical rectangular prisms were printed. When printing was completed, the object was not immediately removed as approximately one hour is required for cool-down.

### 2.2. Powder Characterization

Bulk samples of virgin and used powder were analyzed through inductively coupled plasma optical emission spectroscopy (ICP-OES), scanning electron microscopy (SEM), and particle size distribution (PSD) analysis to determine any significant differences in the size, shape, and elemental composition of the particles. Used powder was gathered from the machine chamber, sifted, and stored for re-use by the facility. Thereafter, it could be collected for analysis whereas the virgin powder was collected from an unused container. The ICP-OES was used to determine the elemental composition of the powder by providing the percentage of the various constituents (Al, Co, Cr, Cu, Fe, Mn, Mo, Ni, Nb, Si, Ta, Ti, and Zn) and was conducted by a nationally accredited laboratory. Both the SEM and PSD analyses were conducted at the North-West University, South Africa. The SEM was conducted using the Phenom pro-desktop Scanning Electron Microscope (Phenom PRO Desktop SEM, Phenom-World B., Eindhoven, The Netherlands). Adhesive carbon strips were used to collect powder for analysis; the strip was placed inside the instrument and observed at 10 kV magnification to visualize particles. PSD analysis was conducted using the Malvern Morphologi G3 system (Malvern Panalytical Ltd., Malvern, UK). About 5 mm^3^ of the sample was dispersed onto the sample plate by the instrument’s dispersion unit. The instrument captures particles by automated scanning and provides morphological information—size distribution, circularity, convexity, and elongation. A single sample of virgin and used powder was analyzed using ICP-OES and SEM whilst PSD analysis of virgin and used powder samples were analyzed in triplicate. 

### 2.3. Particle Emissions

Particle counters were placed as close as possible to the machine, within 1.5 m, to capture the particle number concentration (p/cm^3^) over time in the area. Particle counters included: TSI P-Trak^®^ Ultrafine particle counter model 8525 (TSI Inc., Shoreview, MN, USA) and the Grimm Portable Laser Aerosol Spectrometer model 11-A (GRIMM Aerosol Technik GmbH & Co., Muldestausee, Germany). The particle size range of the instruments differs as follows: the P-Trak^®^ detects the smallest range of particles from 0.02 to 1 µm while the Grimm has 31 channels, detecting particles from 0.25 to 32 µm. The particle counters were set to measure at 60-s intervals for the entire duration of the AM process. The background concentration was determined at the beginning of each day of sampling, and measured for a duration of ten minutes at the same position as particle emission measurements, as close as possible to the machine. Background measurements were conducted before the AM process started but whilst unrelated tasks in the area were conducted, to observe particle emissions solely from AM above this baseline. All instruments were zero-checked before each session. The emission rate (ER) of particles was calculated using the following equation [31,32]:(1)$$\mathrm{ER}\;(p/\min)=V\;\cdot \;\left[\frac{C_{\mathrm{peak}}-C_{\mathrm{out}\;}}{\text{∆}t}+\overset{-}{\mathrm{AER}+k}\cdot {\overset{-}{C}}_{\mathrm{in}}-\mathrm{AER}\;\cdot \;C_{\mathrm{out}}\right]$$
where V is the room volume (m^3^), C_peak_ is the peak particle number concentration during the AM phase (p/cm^3^), C_out_ is the average outdoor particle number concentration (p/cm^3^), C_in_ is the average indoor particle number concentration for the duration of the AM phase (p/cm^3^), ∆t is the time difference between C_peak_ and C_out_ (min), AER is the air exchange rate of the room (air exchange/h), and k is the particulate loss rate due to surface deposition. The air exchange rate was calculated for each room used during the process using the tracer gas decay method, based on the American Society for Testing and Materials (ASTM) methods ASTM E741-11 (2017) and ASTM D6245-07 (2012) [33,34]. Carbon dioxide was injected into the room while data loggers (Kimo KISTOCK KCC 320, Kimo Instruments, Montpon, France) measured the decay over time. This procedure was repeated three times in each room. The k-value for an indoor environment, 1/h, was used [32].

### 2.4. Personal Exposure

To quantify the personal exposure, the two AM operators wore two samplers for the duration of each AM process (including the pre-processing, processing, and post-processing phases). An Institute of Occupational Medicine (IOM) multi-dust sampler was used, containing a foam insert and a 25 mm mixed cellulose ester (MCE) filter. The IOM was used with an aspiration pump (Gilian GilAir Plus, Sensidyne, LP, USA), calibrated to 2 L per minute (L/min). The second sampler used was the Nanozen DustCount^®^ model 9000-Z1 (Nanozen Industries Inc., Vancouver, BC, Canada). This instrument measured real-time particle number concentration, used with a PM10 impactor, to detect particles from 0.3 to 20.8 µm in size. Personal exposure sampling was conducted following the guidelines provided in Health and Safety Executive (HSE) Method MDHS 14/4. The foam inserts and filters were transported to a nationally accredited laboratory where they underwent gravimetric and ICP-OES analysis post-measurement in order to determine the mass concentration and composition of the powder to which AM operators are exposed. The gravimetric analysis determined the mass of particles on each sample and was conducted based on MDHS Method 14, National Institute for Occupational Safety and Health (NIOSH) Method 0500, 0600, and GME 16/2/3/2/3 by the laboratory. After gravimetric analysis, the laboratory split each filter and foam to undergo ICP-OES and analysis for the presence of hexavalent chromium, respectively. The ICP-OES analysis tested for metals (Al, Cd, Co, Cr, Cu, Fe, Mn, Mo, Ni, Pb, Sn, Ti, V, and Zn) and was conducted based on NIOSH Method 7303. Lastly, the laboratory analyzed the filters and foam for hexavalent chromium according to NIOSH Method 7600. Field blanks were analyzed together with samples to establish any contamination during the handling of samples while the laboratory used in-house media for blank correction. The gravimetric and metal ICP-OES data were used together with the volume of air sampled and the duration of each process to calculate the TWA exposure concentration for AM operators. Mass values that were below the limit of quantification were substituted by the limit of detection (LOD) mass value divided by the square root of two.

### 2.5. Data Analysis

Data were analyzed using GraphPad software (GraphPad Prism version 8.0.1, GraphPad Software Inc., San Diego, CA, USA). Powder characterization is represented by ICP-OES (%), PSD (µm ± SD) data as well as SEM images. Particle emissions are reported as particle number concentration (p/cm^3^) and calculated emission rates (p/min) whilst personal exposure is indicated by TWA values (µg/m^3^). Personal exposure mass values below the LOD were substituted with a constant value to obtain estimated exposure. Particle emission rate data were tested for normality, however, the data were not normally distributed and, therefore, non-parametric tests (Kruskal–Wallis H test and Mann–Whitney U test) were used to determine the variation. During analyses, *p*-values ≤ 0.05 were considered statistically significant. SEM images were analyzed using image processing and analysis software, namely, ImageJ (ImageJ.JS version 0.5.5, National Institute of Health, Bethesda, MD, USA), to measure particle diameter.

### 2.6. Ethics Approval

The two AM operators who work with the DED machines at the facility were invited to take part in this study, participation was therefore voluntary, and participants provided informed consent. This study was approved by the Health Research Ethics Committee of the North-West University, South Africa (NWU-00249-21-A1).

## 3. Results

### 3.1. Powder Characterization

ICP-OES analysis performed on the virgin and used inconel powder established the metal composition thereof (Table 2). The main metal constituents of inconel were as follows: in the virgin powder nickel (50.20%), cobalt (0.10%), and chromium (20.70%) were present while in the used powder the percentages were marginally different: nickel (50.80%), cobalt (0.18%), and chromium (20.00%). All the measured metals were well within the range provided in the safety data sheet (SDS), except for tantalum (0.06%) which was lower than the stated range of 4.75 to 5.50%. The constituent metal percentages of the virgin and used powder did not differ significantly, with differences of <1%.

The PSD analysis of the virgin and used inconel powder (Table 3) indicated a mean particle diameter of 10.96 ± 3.01 µm and 10.31 ± 1.32 µm, respectively (*p* = 0.526). The 10th and 50th percentiles of particle size for the virgin and used powder were 0.22 ± 0.00 µm and 0.25 ± 0.01 µm, respectively, while the 90th percentile was 56.29 ± 4.31 µm for the virgin and 57.89 ± 4.74 µm for the used powder. The detection limit of the PSD analysis was 0.22 µm, therefore, it is possible that smaller particles were present but not measurable. According to the manufacturer, particles should not be >45 µm, however, as depicted by the particle size distribution analysis results, 10% of the particles were >56.29 ± 4.31 µm in size. The circularity is determined by the proximity of a particle to a perfect circle; the virgin and used powder both have middling circularity of ~0.55, with no significant difference between them (*p* = 0.275). The convexity of a particle refers to the surface smoothness or roughness of a particle; both virgin and used powder had a relatively high convexity of 0.949 ± 0.003 and 0.950 ± 0.005 respectively (*p* = 0.606). The elongation of a particle refers to the width-to-length ratio; the virgin powder’s elongation was 0.326 ± 0.005 while the used powder was 0.323 ± 0.006 (*p* = 0.338) (see Appendix A for clarification, Appendix A).

The SEM images further confirmed the findings of the PSD analysis with similar appearing particles observed in virgin and used powder (Figure 1). Virgin and used powder particles with diameters ranging between 35.0 and 123.0 µm were observed in the images. Most particles appeared to have an almost spherical shape, with a significant number of satellite particles attached to their surfaces visible. The satellite particles ranged from as small as 0.5 µm to 20.0 µm in size.

### 3.2. Particle Emissions

Table 4 and Table 5 provide an overview of the particle number concentration measured during the three phases of AM for the enclosed LENS machine and open-table LMD machine. The duration of each phase is indicated. While it is important to note during the LENS processing phase the machine door was opened every 30 min and the short duration of both pre- and post-processing phases may have affected the particle number concentrations. The particle number concentration data were divided into ranges that correlate with the health-related size fractions (0.02–1 µm; 0.25–1 µm; 1–4 µm; 4–10 µm; >10 µm). There was a notable difference between particles < 1 µm measured by the two instruments. This is due to the difference in the smaller detection size. Therefore, particles measured by the P-Trak^®^ are assumed to be predominantly between 0.02 and 0.25 µm in size, as the same measurements by the Grimm yielded much lower particle number concentrations.

The process of DED AM caused an increase in the area particle number concentration, especially for submicron particles (<1 µm). Particles > 1 µm in size displayed particle number concentrations of <1.0 × 10^3^ p/cm^3^ during all the phases of AM for both the DED machines. During the AM process using the enclosed LENS machine, the particle number concentration for the smaller particle size range (0.02 to 1 µm) was higher with the maximum particle number concentrations for the pre-processing, processing, and post-processing 1.3 × 10^5^, 9.7 × 10^4^, and 1.3 × 10^5^ p/cm^3^, respectively. The particle number concentration for particles 0.25 to >10 µm in size was relatively low, only increasing slightly above the background concentration, while there was a clear peak during the post-processing phase (3.7 × 10^3^ p/cm^3^). During the process using the open-table LMD machine, the smaller particle range (0.02 to 1 µm) displayed high maximum particle number concentrations of 1.4 × 10^4^, 1.7 × 10^5^, and 1.6 × 10^4^ p/cm^3^ for the pre-processing, processing, and post-processing phases, respectively. Unlike with the LENS machine where post-processing displayed the peak concentration, with the open-table LMD process the peak concentration was during the processing phase. During the LMD process for particles 0.25 to >10 µm in size, the peak particle number concentration was observed during the post-processing phase (2.4 × 10^3^ p/cm^3^).

The AER, determined by using the tracer gas decay method (Appendix A, Appendix A), was as follows: 23 in the workshop, 24 in the AM laboratory, and 17 in the post-processing area with natural ventilation, while it was 31 with the extraction ventilation switched on. As seen in Figure 2, the particle emission rates vary between particle size ranges. Submicron particles had the highest emission rate during all three phases of AM. For particles 0.02 to 1 µm in size, during the LENS process, the pre-processing phase had the highest particle emission rate (2.8 × 10^9^ p/min) while during the LMD process, the processing phase emitted the highest rate of particles (4.8 × 10^8^ p/min). For the larger particle range, from 1 to > 10 µm in size, during the LENS process the overall highest particle emission rate was during post-processing (6.5 × 10^5^ p/min), while during the LMD process, it was higher during the processing phase (5.4 × 10^5^ p/min). There was no significant difference in the particle emission rate between the three phases of AM or between the two different DED machines (all *p* values > 0.05). However, there were significant differences in the particle emission rates between the particle size ranges (*p* values ≤ 0.05). The AER had a substantial effect on the particle number concentration, as seen in Figure 3. There was a significant difference in particle number concentration when the extraction ventilation was switched on during the second cycle. The extraction ventilation lowered the peak particle number concentration by ~3.0 × 10^3^ p/cm^3^.

### 3.3. Personal Exposure

Personal respiratory exposure was conducted for an average duration of 126 ± 14 min and 68 ± 19 min for the LENS and LMD processes, respectively. The inhalable fraction of personal exposure data was used to calculate the estimated TWA exposure concentration for metals of health concern over the full AM process, including all three the phases of AM–pre-processing, processing, and post-processing. The TWA exposure indicates exposure experienced over eight-hours; however, the measurement duration was well below this time-frame. For the remainder of the eight-hour duration, exposure was assumed to be zero, as typically at this facility, AM operators conduct research in an office setting and only perform the AM process as required.

The majority of the measured metal exposure was below the LOD—79 % and 51% of the measured values were below the LOD for the LENS and LMD processes, respectively. The detected metals for the LENS process included chromium, copper, iron, molybdenum, nickel, lead, and titanium, whereas during the LMD process aluminum, chromium, copper, iron, nickel, lead, titanium, and zinc were detected. Therefore, the calculation was used to imitate the worst-case exposure scenario at this facility. Data for the respirable fraction were not included as it was insignificant due to the fact that it cannot be compared to any standard or legislative exposure limits. As seen in Table 6, the estimated average TWA exposure concentration calculated for the inhalable fraction of each metal was low, with all metal exposure being <1% of their respective South African TWA OEL concentrations, with the exception of hexavalent chromium which was <5%. Despite not being mentioned in the SDS, lead was detected during both the LENS and LMD processes (0.003–0.69 µg/m^3^), while tin was also detected during the LMD process (0.008–0.95 µg/m^3^). This is likely due to sandblasting in the general post-processing area, where residue from previous tasks as well as the abrasive used may have influenced samples. Samples were tested for the presence of hexavalent chromium and all measurements were below the detection limit.

Figure 4 displays the total particle number concentration of AM operators’ personal exposure during the process. The most prominent size fraction throughout the AM process was 0.375 µm (Appendix A, Appendix A). During the enclosed LENS process, a gradual increase can be seen from the beginning of the pre-processing phase up to the post-processing phase, with some small peaks during the processing phase which are likely due to the opening of the machine chamber. Clear peaks can be observed during the post-processing phase when the operators performed sandblasting. During the open-table LMD process, the particle number concentration was relatively stable for the duration of the process, except for the clear peaks linked to sandblasting during the post-processing phase.

The sandblasting was performed in an enclosed chamber, however, operators are at close range and are required to open the chamber periodically to inspect the progress of the object during this process. As illustrated by Figure 4, there were two AM operators, and the primary operator carried out the majority of the AM processes, including cleaning and setting up of the AM machine, being in close proximity to the machine, while the secondary operator observed the primary operator and performed most of the post-processing tasks, i.e., sandblasting.

When comparing the real-time particle number concentration measured by the DustCount^®^ for personal exposure monitoring to that of the P-Trak^®^ and Grimm used for the area monitoring, on average, the personal exposure was ~10^2^ p/cm^3^ lower than that of the area monitoring. This is due to the area monitoring consistently being measured within a meter of the process while the personal exposure depended on the AM operator activity. Nevertheless, the peaks observed in Figure 4 are consistent with the peaks in the area monitoring, except during the LMD process. The particle number concentration displayed clear peaks during the processing phase of open-table LMD that were not observed in the personal exposure figures. This, again, can be attributed to the fact that the AM operators were performing different tasks while the automated processing was taking place.

## 4. Discussion

### 4.1. Powder Characterization

Powder characterization plays a vital part in determining the health risks associated with the AM process; particle size and shape both have substantial effects on the route of exposure as well as where it will deposit in the lungs. The chemical and physical composition of the inconel powder did not change significantly after being used, as reported previously by Hann (2016) [23]. The lack of variation between virgin and used inconel can be due to the powder only being re-used once at this specific facility, as well as the specific powder, with characteristics such as high-temperature strength. There was no change in the constituent percentage of more than 1% for any of the metals.

Particles smaller than 100 µm can be inhaled (50% cut-point of 100 µm). The inhalable fraction refers to all particles that are inhaled and may be deposited anywhere along the respiratory tract while the respirable fraction refers to particles that reach the deepest parts of the lung, known as the gas-exchange region. The respirable fraction refers to particles between 3 and 5 µm (50% cut-point of 4 µm) [36]. With 50% of powder particles below 0.25 µm and 90% below ~55 µm, there is a significant chance of inhalation of both virgin and used inconel powder, whilst also containing metals of concern to human health such as respiratory sensitizers (Co) and carcinogens (Ni). Powder particles used in AM are typically spherical, created by a process called gas-atomization. However, during this process smaller particles, namely satellite particles, may attach to the manufactured particles [8]. Additionally, during the process of AM particles may agglomerate creating clusters. As observed through the SEM images (Figure 1), fused and satellite particles are the cause of the decreased circularity and increased elongation factor in this case. It is important to note that the irregular shape of these particles does affect the deposition in the lungs [8].

### 4.2. Particle Emissions

The measured particle number concentration for submicron particles during the AM process ranged between 10^2^ and 10^5^ p/cm^3^, which is slightly lower than previous studies (10^4^–10^6^ p/cm^3^) [1,22,24]. A general trend observed is the significantly higher particle number concentrations for submicron particles, <1 µm. Compared to the feedstock powder, with a mean diameter of 10.96 ± 3.01 µm, the emitted particles are notably smaller. The particle size range that was significant throughout the process was 0.02 to 0.25 µm, which correlates with previous studies findings that 0.05 to 0.3 was the peak size range during DED AM [1,22,24]. When comparing the peak particle number concentration of the processing phase, there is a clear difference between the enclosed LENS (9.7 × 10^4^ p/cm^3^) and open-table LMD (1.7 × 10^5^ p/cm^3^) machines, with the open-table machine having a higher maximum particle number concentration during this phase. This also correlates with a study by Kugler et al. (2021) where the particle number concentration was around 9.3 × 10^5^ p/cm^3^ next to an open-table DED machine [24]. The particle emission findings were similar to Oddone et al. (2022), who also investigated an enclosed DED machine, where a gradual increase was observed in the submicron particles (0.3–0.5 µm) during the processing phase [27]. Although the open-table machine displayed higher particle number concentrations, the enclosed machine had a noteworthy increase above the background, indicating that particles were still being emitted during the enclosed processing phase. This may be attributed to the regular opening of the chamber door, releasing particles. When comparing DED AM to other metal AM process categories, Oddone et al. (2022) found that the particle number concentration for particles in the 0.3 µm size fraction during the powder bed fusion (PBF) process had a two times higher particle number concentration than DED [27]. In contrast, Graff et al. (2017) measured ~1.6 × 10^4^ p/cm^3^ for particles <0.3 µm during cleaning of the PBF AM machine, which is comparable to the average particle number concentration measured for particles between 0.02 and 1 µm while cleaning during the pre-processing phase of DED in this study (7.7 × 10^3^–3.1 × 10^4^ p/cm^3^) [6]. It should be noted that the before mentioned studies used different optical counters to measure particle number concentration, with differing particle size ranges which affect the particle number concentration. The importance of not only implementing control measures but also providing the proper instruction and training on how to use them was evident from AER measurements. There was a substantial decrease in the particle number concentration when sandblasting was performed while using extraction ventilation.

Currently, there are no published studies that have calculated a particle emission rate for metal AM. The particle emission rate provides a better indication of particles emitted by the AM process as it considers various parameters such as the air exchange rate, volume of the room, and the background particle number concentration. It was noted that in general, submicron particles had higher particle emission rates over the entire process of AM. There was no significant difference found between the three phases of AM or the two DED machines used, however, there was a significant difference in particle emission rate between the various size ranges. The peak particle emission rate during the enclosed LENS process was observed during the pre-processing phase for submicron particles and the post-processing phase for particles >1 µm in size. Peak particle emission rates were observed during the processing phase for all particle size ranges during the open-table LMD process. The particle emission rate calculations indicated the importance of an enclosed system. When compared to other AM process categories such as material extrusion, with particle emission rates ranging from 10^9^ up to 10^12^ p/min, the particle emission rate of DED calculated in this study falls within the lower range with a maximum of 2.8 × 10^9^ p/min [31,32,37].

### 4.3. Personal Exposure

The gravimetric analysis of personal exposure was expected to be low, seeing as a significant percentage of the airborne particles in the area of DED AM are in the submicron range. Mass-based analysis does not capture the essence of exposure to submicron particles, however, it was important to investigate and confirm this. Most metal exposures were below the detection limit and the estimated TWA exposure concentration calculations indicated that exposure is not of any concern, including exposure to the three main metals—nickel, chromium, and cobalt. No hexavalent chromium was detected. Particle number concentration data shows that a great deal of exposure is linked to the post-processing phase, where clear peaks can be observed during the sandblasting of parts. Additionally, a slightly higher particle number concentration for the primary operator can be attributed to the close proximity to tasks and the machine during AM processes. The personal exposure data confirms that traditional gravimetric exposure monitoring alone is not effective in providing an overview of AM operator exposure. Although no adverse health effects are expected considering the inhalation of metal constituents is low, other routes of exposure could contribute to total uptake. Additionally, there are metal constituents considered carcinogenic and, therefore, it is imperative to keep AM operator exposure as low as reasonably practicable with appropriate control measures, as executed at this facility.

A few limitations were experienced during the execution of this study, including time restraints and unrelated tasks occurring in the vicinity of the AM process. However, this is to be expected in real-world workplace conditions. There is potential for future studies in AM, considering the various process categories, technologies, and available feedstock materials. On completion of this study, recommendations to future studies are studies repeating this research using other metal powders as feedstock material and more studies calculating the particle emission rate for different AM process categories—especially metal AM.

## 5. Conclusions

This study evaluated the emissions and exposure associated with DED AM using an inconel powder, being the first to calculate a particle emission rate in this scenario. Valuable information has been gained to assist in understanding and subsequently assessing and controlling AM operator exposure during the process of DED AM, including the particle emission rate during the process of DED which ranges between 10^6^ and 10^9^ p/min for submicron particles. This is especially relevant to peak emissions identified during the post-processing phase which allows targeted mitigation of the source such as process enclosure. The findings of this study correlate well with previous related metal AM studies, and together these should be a benchmark to encourage future studies aiming to quantify emissions in the continuously advancing field of AM.

## Figures and Tables

**Figure 1 ijerph-20-06206-f001:**
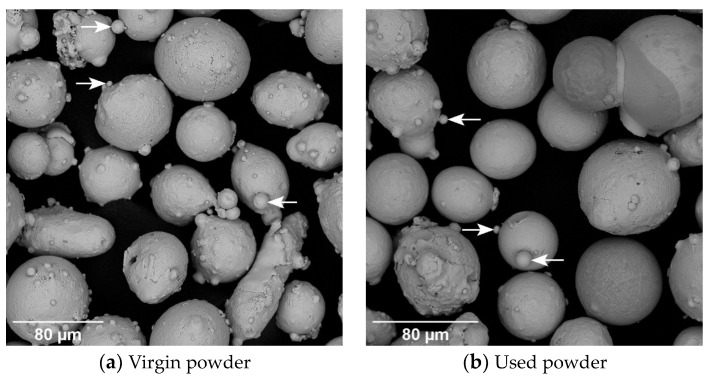
SEM images of virgin and used powder (arrows on the figure indicate examples of satellite particles; scale indicated on images; magnification of images are (**a**) 900× and (**b**) 1050×, respectively).

**Figure 2 ijerph-20-06206-f002:**
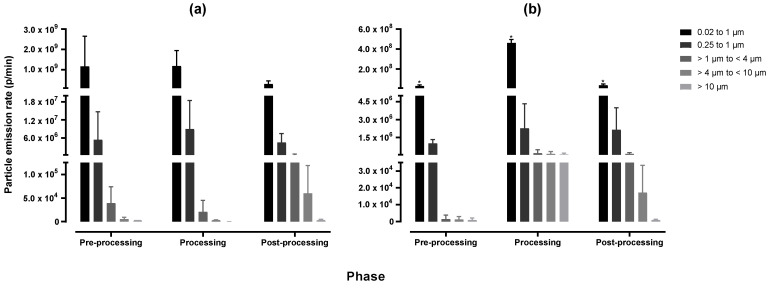
Average particle emission rate (p/min) for different particle size ranges over three cycles, calculated for (**a**) LENS and (**b**) LMD (note difference in y-axis scale and AER; * number of cycles = 2).

**Figure 3 ijerph-20-06206-f003:**
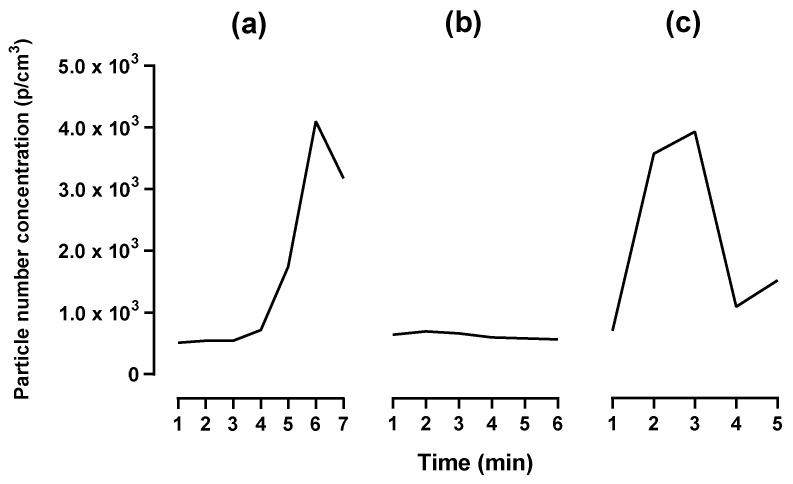
Particle number concentration measured during the post-processing phase of the three repeated cycles of LENS (Cycle (**a**) and (**c**) was performed with natural ventilation whereas during cycle (**b**), extraction ventilation was used).

**Figure 4 ijerph-20-06206-f004:**
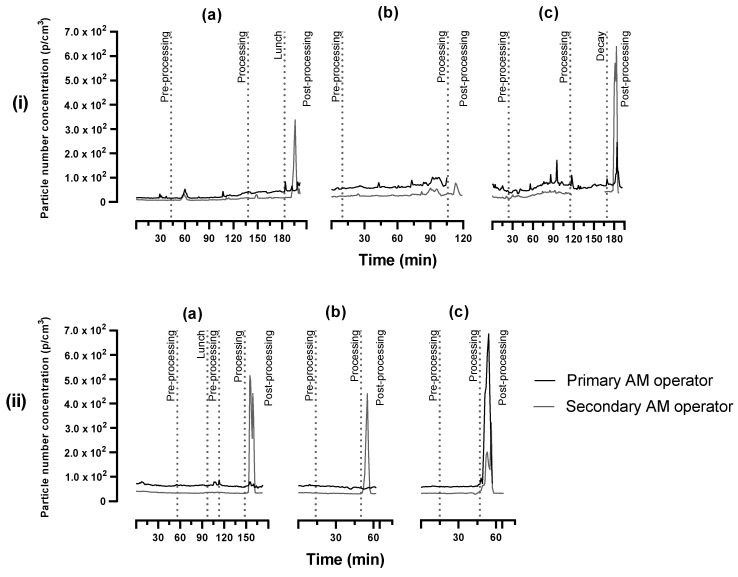
Total particle number concentration (p/cm^3^) of AM operators’ personal exposure during the (**i**) LENS and (**ii**) LMD processes for (**a**) cycle 1, (**b**) cycle 2, and (**c**) cycle 3.

**Table 1 ijerph-20-06206-t001:** Summary of facility layout, activities performed by AM operators, and implemented control measures.

Location Description	Simple Illustration		Activities	Control Measures
General workshop Various tasks are performed in this area, which includes AM in a dedicated area. The enclosed LENS machine is situated here.	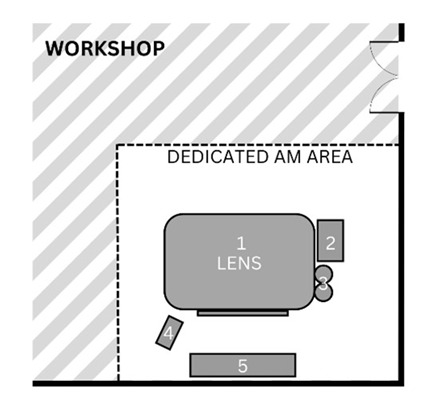	1 Enclosed LENS machine 2 Workstation 3 Machine reservoir 4 Machine control panel 5 Desk	Pre-processing AM operators prepare the machine by cleaning and filling the reservoir (2, 3) and setting up machine parameters (4). Processing Automated building of object by LENS machine (1). AM operators complete other tasks, seated at desk (5).	The workshop makes use of natural ventilation through a large door as well as windows. The machine has a safety glass panel through which the process can be monitored, protecting eyes from the laser. Safety notices are displayed around the machine as well as SOPs. Protective clothing and safety shoes were worn by the AM operators (no respirators or gloves were worn when handling the powder).
AM laboratory This laboratory is used solely for AM. The open-table LMD machine, consisting of a table and a robotic arm, is situated here.	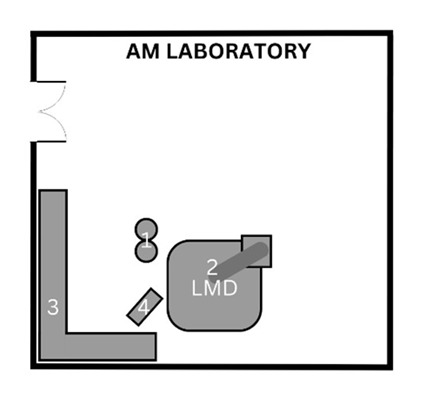	1 Machine reservoir 2 Open-table LMD machine 3 Workstation 4 Machine control panel	Pre-processing AM operators prepare the machine by cleaning and filling the reservoir (1, 2) and setting up machine parameters (4). Processing Automated building of object by LMD machine (2). AM operators complete other tasks, seated at desk (3).	The AM laboratory makes use of natural ventilation. Safety notices are displayed at the entrance with SOPs next to the machine. Eye protection, protective clothing and safety shoes were worn by the AM operators for the duration of the process while respirators were worn when handling powder (no gloves were worn when handling the powder).
Post-processing area The post-processing area has various finishing machinery and workstations that are used for a variety of post-processing tasks, including but not limited to AM.	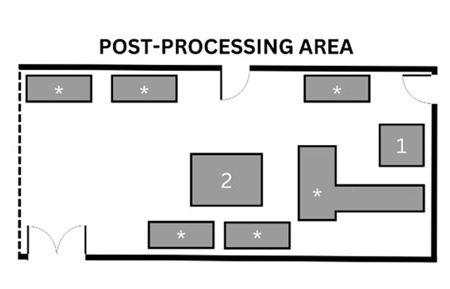	1 Enclosed sandblasting station 2 Workstation * Various finishing machinery	Post-processing AM operators sandblast printed object (1).	There is an extraction ventilation system located above the sandblasting station, however, this was not used consistently. The sandblasting station is enclosed. The AM operators wore protective gloves whilst sandblasting.

**Table 2 ijerph-20-06206-t002:** Metal composition of virgin and used inconel powder compared to the SDS.

Powder	Elemental Composition (%)												
	Al	Co	Cr	Cu	Fe	Mn	Mo	Ni	Nb	Ta	Ti	Si	Zn
Virgin	0.49	0.10	20.70	0.03	18.80	< 0.01	2.87	50.20	5.36	0.06	1.01	0.07	0.20
Used	0.53	0.18	20.00	0.05	18.20	0.03	3.03	50.80	5.65	0.06	1.04	0.09	0.21
SDS	0.20–0.80	1.00 max	17.00–21.00	0.30 max	15.00–21.00	0.35 max	2.80–3.30	50.00–55.00	4.75–5.50	4.75–5.50	0.65–1.15	0.35 max	-

Legend: Al—Aluminum; Co—Cobalt; Cr—Chromium; Cu—Copper; Fe—Iron; Mn—Manganese; Mo—Molybdenum; Ni—Nickel; Nb—Niobium; Ta—Tantalum; Ti—Titanium; SDS—Safety data sheet; Si—Silicone; Zn—Zinc; max—maximum; -Not listed in SDS [14].

**Table 3 ijerph-20-06206-t003:** Particle size distribution and shape analysis of virgin and used inconel powder (mean ± SD) compared to the SDS.

Powder	SDS (µm)	Mean Diameter (µm)		Particle Size Distribution (µm)						Particle Shape Analysis					
				d [0.1]		d [0.5]		d [0.9]		Circularity		Convexity		Elongation	
Virgin	≤45	10.96	±3.01	0.22	±0.00 *	0.25	±0.01	56.29	±4.31	0.552	±0.019	0.949	±0.003	0.326	±0.005
Used	-	10.31	±1.32	0.22	±0.00 *	0.25	±0.01	57.89	±4.74	0.558	±0.014	0.950	±0.005	0.323	±0.006

Legend: d [0.1] 10% of the particles are smaller than the stated diameter; d [0.5] 50% of the particles are smaller than the stated diameter; d [0.9] 90% of the particles are smaller than the stated diameter; -Not indicated in SDS [35]; * 0.22 µm = lower detection limit.

**Table 4 ijerph-20-06206-t004:** Average particle number concentration (p/cm^3^) as measured over three cycles of the enclosed LENS process.

Phase Duration		Particle Size Range				
		0.02–1 µm *	0.25–1 µm	1–4 µm	4–10 µm	>10 µm
Background	Mean	9.2 × 10^3^	3.6 × 10^2^	1.3 × 10^0^	1.0 × 10^−1^	3.7 × 10^−3^
	SD	1.7 × 10^3^	1.5 × 10^1^	8.5 × 10^−1^	5.6 × 10^−2^	2.9 × 10^−3^
	Max	1.7 × 10^4^	5.2 × 10^2^	7.0 × 10^0^	4.6 × 10^−1^	1.4 × 10^−2^
Pre-processing 25 ± 15.49 min	Mean	3.1 × 10^4^	4.1 × 10^2^	1.8 × 10^0^	1.8 × 10^−1^	1.0 × 10^−2^
	SD	9.1 × 10^3^	1.9 × 10^1^	5.7 × 10^−1^	7.9 × 10^−2^	8.6 × 10^−3^
	Max	1.3 × 10^5^	5.9 × 10^2^	6.0 × 10^0^	6.8 × 10^−1^	3.9 × 10^−2^
Processing 94 ± 1.25 min	Mean	3.2 × 10^4^	4.9 × 10^2^	1.4 × 10^0^	1.2 × 10^−1^	6.0 × 10^−3^
	SD	1.3 × 10^4^	1.2 × 10^2^	6.8 × 10^−1^	5.9 × 10^−2^	4.2 × 10^−3^
	Max	9.7 × 10^4^	9.2 × 10^2^	6.8 × 10^0^	7.9 × 10^−1^	4.4 × 10^−2^
Post-processing 9 ± 2.63 min	Mean	6.4 × 10^4^	1.4 × 10^3^	8.4 × 10^1^	1.3 × 10^1^	5.7 × 10^−1^
	SD	1.0 × 10^4^	5.7 × 10^2^	4.6 × 10^1^	8.4 × 10^0^	4.3 × 10^−1^
	Max	1.3 × 10^5^	3.7 × 10^3^	3.3 × 10^2^	5.8 × 10^1^	3.0 × 10^0^

***** Particle number concentration as measured with P-Trak^®^, with all other data measured by Grimm.

**Table 5 ijerph-20-06206-t005:** Average particle number concentration (p/cm^3^) as measured over three cycles of the open-table LMD process.

Phase Duration		Particle Size Range				
		0.02–1 µm *	0.25–1 µm	1–4 µm	4–10 µm	>10 µm
Background	Mean	4.7 × 10^3^	4.5 × 10^2^	1.9 × 10^0^	1.5 × 10^−1^	7.1 × 10^−3^
	SD	4.0 × 10^2^	6.0 × 10^0^	1.5 × 10^−1^	2.7 × 10^−2^	5.4 × 10^−3^
	Max	5.4 × 10^3^	4.6 × 10^2^	2.1 × 10^0^	2.0 × 10^−1^	2.0 × 10^−2^
Pre-processing 25 ± 15.49 min	Mean	7.7 × 10^3^	5.5 × 10^2^	1.9 × 10^0^	3.0 × 10^−1^	1.0 × 10^−1^
	SD	8.6 × 10^2^	8.0 × 10^1^	1.5 × 10^0^	6.0 × 10^−1^	4.1 × 10^−1^
	Max	1.4 × 10^4^	1.5 × 10^3^	2.0 × 10^1^	8.2 × 10^0^	5.9 × 10^0^
Processing 32 ± 0.94 min	Mean	5.9 × 10^4^	7.0 × 10^2^	2.3 × 10^1^	1.4 × 10^1^	8.3 × 10^0^
	SD	2.2 × 10^4^	1.5 × 10^2^	3.0 × 10^1^	2.0 × 10^1^	1.2 × 10^1^
	Max	1.7 × 10^5^	1.4 × 10^3^	2.2 × 10^2^	1.5 × 10^2^	1.1 × 10^2^
Post-processing 9 ± 2.63 min	Mean	1.3 × 10^4^	9.1 × 10^2^	2.8 × 10^1^	4.0 × 10^0^	1.7 × 10^−1^
	SD	1.3 × 10^3^	4.0 × 10^2^	3.2 × 10^1^	4.0 × 10^0^	1.7 × 10^−1^
	Max	1.6 × 10^4^	2.4 × 10^3^	1.6 × 10^2^	1.7 × 10^1^	9.6 × 10^−1^

***** Particle number concentration as measured with P-Trak^®^, with all other data measured by Grimm.

**Table 6 ijerph-20-06206-t006:** Estimated average inhalable fraction TWA exposure concentration (µg/m^3^ ± SD) compared to the respective South African eight-hour TWA OELs of metals [30].

Metal		TWA (µg/m^3^)				Inhalable Fraction TWA OEL(µg/m^3^)
		LENS		LMD		
Al	Aluminium	0.0021	±0.0019 *	0.0075	±0.0075	-
Co	Cobalt	0.0025	±0.0023 *	0.0009	±0.0000 *	40.0000
Cr	Chromium	0.0016	±0.0013	0.0007	±0.0001	1000.0000
Cr(VI)	Hexavalent chromium	0.0156	±0.0157 *	0.0002	±0.0000 *	0.4000
Cu	Copper	0.0019	±0.0018	0.0011	±0.0005	2000.0000
Fe	Iron	0.0221	±0.0204	0.0066	±0.0001	10,000.0000
Mn	Manganese	0.0021	±0.0019 *	0.0007	±0.0000 *	-
Mo	Molybdenum	0.0014	±0.0011	0.0007	±0.0000 *	10,000.0000
Ni	Nickel	0.0018	±0.0016	0.0017	±0.0007	1000.0000
Pb	Lead	0.0020	±0.0019	0.0230	±0.0315	150.0000
Ti	Titanium	0.0043	±0.0051	0.0023	±0.0000	10,000.0000
Zn	Zinc	0.0041	±0.0038 *	0.0048	±0.0042	-

***** All values below LOD.

## Data Availability

Data is contained within the article or Appendix A.

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
