# Peer review of "Emissions and Exposures Associated with the Use of an Inconel Powder during Directed Energy Deposition Additive Manufacturing"

_ijerph, 2023, doi:10.3390/ijerph20136206_

Round 1
Reviewer 1 Report
The article presents original data about exposure and emission assessment of metal additive manufacturing (MAM) using DED process. The method and results are original in a field promising important transformations regarding the way we produce metal parts worldwide. The MAM processes are evolving rapidly and knowledges about exposure are of interest. The article is well writed and brings pertinent knowledges and methods regarding risks related to MAM. For these reasons, we recommend a minor revision to bring some details relating to method and results section.
The minor revisions recommended are listed below:
1) abstract page 1, line 11: Detail composition of inconel 718.
2) p.1, l.20: The fact that there is no exposure limits for exposure to nanoparticles should be addressed in the article.
3) p.1, l.39: Is it possible to add a reference after the sentence "AM has seven standard process categories which include DED"?
4) p.2, l.51: Should we read "exposure to of AM operators"?
5) p.2, l.68 and following: the number concentration of the particles are mentioned. Would it be possible to precise the size of the particles? The number concentration variations are often related to the particles size.
6) p.2, l.82: what classification was used for the carcinogenic classification?
7) p.2, l.86: A new paragraph can start with "Considering that..." to highlight the objectives of the paper.
8) p.3, l.115 to 117: "To achieve the objectives of this study and the research conducted by the facility, identical rectangular prisms were printed.": was it requested by the study plan?
9) p.3, l.117 to 118: "When printing was completed, the object was not immediately removed as time is required for cool down.": What was this duration approximatively?
10) p.3, l.119: 2.2 "particle characterization": Wouldn't "powder chararacterization" be more appropriate?
11) p.3, l.126: Please explain what SANAS is.
12) p.4, l.151: Is there any link between "Cout" and "background concentration"?
13) p.4, l.188: Information on the duration of the tasks, the method for observations (video recording, transcribe notes, etc.), and the duration of the measurements are missing. It would have been appropriate to refer to the international recommendations on strategies for assessing nanoparticles exposures.
14) p.4, l.196: We would also like to read here how the data will be presented (PSD, number concentrations, variations of concentrations, ...)
15) p.7, l.269: We need to know how the background was measured (position of the instruments, duration of the recording, co activities, etc.)
16) p.7, l.271: The recording of higher maximal number concentration is normal. Smaller are the particle, higher is the number concentration... Maybe you can say some words about it.
17) p.8, l.302 to 304: Is there any hypothesis to explain these results?
18) p.10, figure 4: The curves are on the text in some cases ("post processing" 2 a b c and 1 c)
19) p.12, l.443: It is definitely a loss not to have a description of the tasks performed by the operators. This would be an input to understanding exposure in relation to other MAM processes, and an input to the literature. After reading the supplementary material, I think the Table S1 should be placed in the article. The remaining supplementary material doesn't appear necessary in my view.
20) p.12, l.454: Please delete "succesfully".
Author Response
Please see attachment for Reviewer 1.

Reviewer 2 Report
Please see the comments attached in this supplement.

Quality of English is ok.
Author Response
Please see the attachment (rebuttal Reviewer 2)

Reviewer 3 Report
This is a very interesting paper that contributes to increase knowledge in area of concern. It is well written and well structured. The research questions are clear and text in consistent.
Anyway, there are a few questions that authors should consider:
Line 30: Authors mention "through the layering and fusing of materials". This applies to metals additive manufacturing (AM) but not to al types of AM.
2. Materials and methods: Why do authors not considered to analyse emitted particles in SEM?
Equation (1): Authors should present the source (reference)
2.2. Particle characterization: Please clarify where used powder was collected
3.3 Personal exposure: Since personal sampling includes the 3 phases of work and post-processing is the more relevant, with higher particle concentration, it is important consider that most of the analysed particles are from that phase. So, being sandblasting referred, particles of the used abrasive will be present and influence the results. Considering that this paper is focused in AM, and particles released during AM process it was important that sampling should be different for specific AM processes (Phase 1 and 2) from the finishing process that is common with other metal processing technologies. Anyway, authors should identify the used abrasive or abrasives in the finishing process.
Table 5: results below LOD/LOQ should be identified.
4. Discussion: Authors do not mention and discuss the differences found in composition between powder and particles collected in air. This is relevant because workers are not exposed to powder (large particles, not airborne or rapidly settled) but to smaller particles that were collected in personal sampling filters.
4.3 Personal exposure: It is not possible to correlate emission and exposure. First, author used gravimetric analysis and recognize that it is not the best way to characterise exposure to ultrafine particles. Second, the equipment used to measure exposure in particle number (Nanozen), only measures above 300 nm2, and in the emission results it is clear that the size fraction between 20 nm and 250 nm is very important. In my opinion, personal exposure results should be analysed with special care and are a limitation in this work. Authors should consider to compare the results only in larger particles (above 300 nm) or take another option.
Line 439 to 441: Considering that particle in nanoscale are emitted, leading to possible exposure of workers, translocation of particles in the body is a concern to workers health. Authors should consider this issue.
Conclusions
It is suggested to the authors to focus on emission rather than exposure. Exposure follows emission so, without emission will be no exposure. Considering that results are more robust in emission than in exposure, and emission control is hierarchically superior to exposure control, recommendation on lower emission rates and/or enclosed processes could be made.
Author Response
Please see the attachment (rebuttal Reviewer 3)

Round 2
Reviewer 2 Report
Thank you for the corrections made. I still have comments to the manuscript.
- Please correct significant figures in the whole manuscript.
- Please include how the values below LOD are handled in the data analysis subchapter.
- "Significant figures corrected in Tables 2, 3 and 6": – Please check this again.
- Table 4 and 5: What does <1.0x100, <1.0x101, <1.0x102 mean? Is it LOD and LOQ?
- Figure 2: Please include how the standard deviation is calculated for n<3 cycles? Please add how many cycles instead of n<3.
- Please revise table 6: How is the standard deviation calculated for all values below LOD? Please correct the significant figures. The high level of certainty reported cannot be correct when it is an estimate of exposure.
Line l.412 and l.413: “Powder particles used in AM are typ- 412 ically circularspherical, created by a process called gas-atomization, however, during this 413…” does not include “additional information regarding values below LOD”.
- Please revise this sentence (L.397-398): “Powder characterization plays a vital part in determining the health risk associated to the AM process, particle size and shape both have substantial effects on the route of exposure as well as where it will deposit in the lungs.” Powder characterisation does not play a vital part in determining health risk. It only applies for airborne particles.
OK
